# Effects of Bacterial and Fungal Inocula on Biomass, Ecophysiology, and Uptake of Metals of *Alyssoides utriculata* (L.) Medik.

**DOI:** 10.3390/plants12030554

**Published:** 2023-01-26

**Authors:** Silvia Priarone, Sara Romeo, Simone Di Piazza, Stefano Rosatto, Mirca Zotti, Mauro Mariotti, Enrica Roccotiello

**Affiliations:** 1Dipartimento di Scienze della Terra, dell’Ambiente e della Vita, Università degli Studi di Genova, Corso Europa 26, 16132 Genova, Italy; 2Agenzia Regionale Protezione dell’Ambiente Ligure, Via Bombrini 8, 16149 Genova, Italy

**Keywords:** Fv/Fm, metals, performance index, PGP

## Abstract

The inoculation of plants with plant-growth-promoting microorganisms (PGPM) (i.e., bacterial and fungal strains) is an emerging approach that helps plants cope with abiotic and biotic stresses. However, knowledge regarding their synergic effects on plants growing in metal-rich soils is limited. Consequently, the aim of this study was to investigate the biomass, ecophysiology, and metal accumulation of the facultative Ni-hyperaccumulator *Alyssoides utriculata* (L.) Medik. inoculated with single or mixed plant-growth-promoting (PGP) bacterial strain *Pseudomonas fluorescens* Migula 1895 (SERP1) and PGP fungal strain *Penicillium ochrochloron* Biourge (SERP03 S) on native serpentine soil (*n* = 20 for each treatment). Photosynthetic efficiency (Fv/Fm) and performance indicators (PI) had the same trends with no significant differences among groups, with Fv/Fms > 1 and PI up to 12. However, the aboveground biomass increased 4–5-fold for single and mixed inoculated plants. The aboveground/belowground dry biomass ratio was higher for plants inoculated with fungi (30), mixed (21), and bacteria (17). The ICP-MS highlighted that single and mixed inocula were able to double the aboveground biomass’ P content. Mn metal accumulation significantly increased with both single and mixed PGP inocula, and Zn accumulation increased only with single PGP inocula, whereas Cu accumulation increased twofold only with mixed PGP inocula, but with a low content. Only Ni metal accumulation approached the hyperaccumulation level (Ni > 1000 mg/kg DW) with all treatments. This study demonstrated the ability of selected single and combined PGP strains to significantly increase plant biomass and plant tolerance of metals present in the substrate, resulting in a higher capacity for Ni accumulation in shoots.

## 1. Introduction

Metal pollution in soil has become a global environmental issue due to the intense increase in industrialization and intensive agricultural activities [1]. Metals are highly toxic and persistent pollutants because they are not biodegradable, and their oxidation state can change, with a half-life longer than 20 years [2]. Metallic pollutants have caused health problems for approximately 10 million people worldwide [3], and they represent 37% of contaminants in European soils and waters [4]. Metal-contaminated soils also act as stressors on plants, causing photosynthetic activity alterations and affecting global ecophysiological performance [5,6,7,8] by inhibiting plant growth and biomass production [9].

Hyperaccumulators are plants able to effectively transfer metals to their shoots [10], reaching a concentration >1% (depending on the metal) [10,11,12]. This ability can be exploited for phytoextraction in metal-rich soils enriched by natural or anthropogenic inputs [13] to concentrate metals in the aboveground biomass [2]. Phytoextraction is a cost-effective and sustainable technology with good public acceptance, although the process does require a significant amount of time to succeed. To improve metal hyperaccumulation, several assisted phytoremediation techniques can be exploited: (i) using chelating agents that increase the uptake and translocation of metals to the shoots [14,15]; (ii) the effective addition of exogenous phytohormones [16,17,18,19,20]; and (iii) inoculation with plant-growth-promoting microorganisms (PGPM), an emerging approach that employs plant growth promoting bacteria (PGPB) and plant-growth-promoting fungi (PGPF) to alleviate metal stress and support metal uptake [21]. However, chelating agents affect plants’ health via stunted growth and leaf chlorosis [14], reduced biomass [14], or increased risk of groundwater pollution via metal mobilization, as in the case of Ethylenediaminetetraacetic Acid (EDTA) [22]. Exogenous phytohormones impact plant yield [16,23,24]. Regarding PGPM, the combined use of fungi and bacteria requires prior evaluation to rule out antagonistic action between strains. It is well known that some fungal strains can produce effective antibacterial secondary metabolites.

Given expanding interest in the PGPM approach, new studies are required to increase knowledge regarding their effects on plants growing in metal-rich soils.

PGPB are directly and indirectly involved in plant growth stimulation. PGPB are directly involved in minerals solubilization, promotion of nutrients uptake, nitrogen fixation, phosphate solubilization, production of siderophores, and secretion of hormones and metabolites [24,25,26]. PGPB are indirectly involved in protection against pathogens [27,28], increased plant resistance to abiotic stresses, such as excess soil metals [29], and reduction of metal toxic effects [24]. Indeed, some PGPB have potential for phytoremediation purposes [30]. Among these, *Pseudomonas fluorescens* complex Migula, 1895 are ubiquitous in rhizospheric microbiota [31] and tolerant to metals [32,33]. These bacterial strains were also able to increase crop yield (e.g., *Solanum tuberosum* L.) [34], growth parameters (e.g., *Pisum sativum* L., up to 45%; *Phaseolus vulgaris* L., up to 20%) [35], and shoot dry biomass (e.g., *Eragrostis tef* (Zucc) Trotter, up to 2.8-fold) [36].

PGPF were more recently investigated because of their key roles in the PGP process [37,38,39]. Similar to PGPB, PGPF can directly and indirectly interact with plants, improving germination rate, photosynthetic efficiency, biomass production, stimulating phytohormones production, etc. [37,40]. Other saprotrophic or mycoparasitic species can act as antagonists of harmful species by modulating the expression of different genes [38]. Until now, most strains that have shown the greatest PGP effect by improving horticultural plants’ (e.g., *Brassica rapa* L., *Saccharum officinarum* L., *Cucumis sativus* L., etc.) dry biomass up to 170% belong to different genera; the most studied strains are *Penicillium* and *Trichoderma*. Moreover, their natural ability to produce siderophores to chelate metals can be fruitfully exploited for bioremediation purposes [39]. For instance, certain *Pseudomonas fluorescens* strains can detoxify organic and inorganic pollutants; *Penicillium ochrochloron* is a metal-resistant fungus highly tolerant to copper and Ni, with metal uptake ability [32,40,41].

Currently, most studies of assisted phytoremediation techniques regarding microorganisms focus primarily on endophytes as mutualistic symbionts of plants that improve plant performance under extreme conditions, such as drought, nitrogen deficiency, salinity, and metal phytotoxicity exposure. Endophyte inoculation has shown potential for plant growth promotion and can increase metal translocation in hyperaccumulator shoots by mitigating stresses from contaminated and naturally metal-rich soils [42]. However, knowledge regarding mixed PGPB and PGPF microbial consortia that support plants at the rhizosphere level is lacking.

We previously identified PGP strains of *Penicillium ochrochloron* and *Pseudomonas fluorescens* isolated from the Ni-hyperaccumulator *Alyssoides utriculata* (L.) Medik. [32] rhizosphere as synergically able to co-grow in a mixed culture, developing a biofilm where the two microorganisms merge and reach their mature stages [41]. Their synergistic behavior suggests their potential use as in vivo microorganism consortia to mitigate metal stress and promote metal uptake for bioremediation purposes [33].

Hence, these strains were employed in this in vivo experiment on metalliferous soil to directly evaluate their potential synergic role in *A. utriculata’s* rhizosphere for phytoremediation purposes. Our study evaluates and compares the effects of co-inoculations and single inoculations of PGP fungal and bacterial strains on *A. utriculata* from an assisted phytoremediation perspective in terms of biomass production, metals and nutrients uptake, and physiological responses.

## 2. Results

### 2.1. Biomass Production

First, data for aboveground and belowground dry biomasses were analysed. Figure 1′s boxplots show the difference between the biomass of plants inoculated with PGPB (fourfold), PGPF (fivefold) and mixed inocula (fivefold) over the control group. The biomass of plants that received mixed inocula were significantly greater compared with the control group’s biomass, as shown by the T-test (Table 1). 

There was a significant difference in biomass production between controls and treated plants (Table 1). The control group had a significantly lower biomass production in comparison with all three treated plant groups (*p* < 0.001 for aboveground and belowground biomass), whereas there was no significant difference in biomass production between the single and mixed inocula.

Analyses concentrated on aboveground biomass, as it was the most relevant for phytoextraction purposes. Furthermore, the aboveground/belowground dry biomass ratio showed that the aboveground biomass was 17-fold in the bacteria group, 21-fold in the control group and the mixed group, and 30-fold in the fungi group.

### 2.2. Heavy Metals and Nutrients Accumulation in the Aboveground Biomass

The accumulation of elements in the samples’ aboveground biomass was quantitatively analyzed via descriptive analysis (Figure 2) and inferential statistical analysis to compare averages (Table 2).

Figure 2 summarizes differences in the aboveground biomass metals accumulation for PGP treatments and the control. Ni, the most absorbed element, accumulated differently with each treatment, as follows: fungi > bacteria > mixed > control. 

An unpaired two-sample T-test summarized the significant differences in absorption of macronutrients, such as P, S, and sometimes N, and micronutrients, such as Mn, Cu, Zn, and B (Table 2). 

No significant differences between the control and treatments were recorded for Ni concentrations (Table 2). However, P concentrations (control, P = 0.14%; fungi, P = 0.18%; bacteria, P = 0.19%; and mixed P = 0.2%) and S concentrations (control, S = 2.18%; fungi, S = 1.63%; mixed, S = 1.67%; and bacteria, S = 1.68%) were significantly different. Interestingly, N concentrations for the mixed and control groups differed (control, N = 3.42%; and mixed, N = 3.11%). Micronutrients, such as Mn (control, Mn = 110.5 mg/kg; mixed, Mn = 208.5 mg/kg; fungi, Mn = 229.8 mg/kg; and bacteria, Mn = 414 mg/kg) and Cu (control, Cu = 6.17 mg/kg; fungi, Cu = 2.61 mg/kg; bacteria, Cu = 3.43 mg/kg; and mixed, Cu = 13.28 mg/kg) were significantly different for all treated plant groups compared with the control. Single inoculum results also differed for Zn (control, Zn = 152.7 mg/kg; bacteria, Zn = 231.6 mg/kg; and fungi, Zn = 244.4 mg/kg) and B (control, B = 42.3 mg/kg; bacteria, B = 35.88 mg/kg; and fungi, B = 32.7 mg/kg) concentrations. The control group had higher S and B values than single and mixed inocula did (except B in the mixed inocula). The control’s N level was significantly higher than the mixed group’s was. The single and mixed inocula always had significantly higher P and Mn concentrations than the control did, whereas there was a higher Zn level only in the single inocula. Cu accumulation was higher in the control than in single inocula and significantly higher in the mixed group.

The macronutrient concentration in the aboveground biomass varied (Appendix A); all plants in each treatment group had similar macronutrient accumulations, whereas the highest concentration variability within groups was for micronutrient accumulation, specifically metals (Appendix A).

Figure 3′s radar chart illustrates metal concentrations in treatment groups’ averaged aboveground biomass; Fe (highest in fungi) and Mn (highest in bacteria) had the widest concentration ranges among treatment groups, whereas Ni concentration was approximately that of hyperaccumulation (Ni > 1000 mg/kg DW) for all treatments. However, Ni and Fe concentrations of plants treated with single and mixed inocula were not significantly different, whereas the difference in Mn concentrations was relevant. 

### 2.3. Photosynthetic Efficiency and Performance Index 

For each plant sample, three measurements were taken on the third and fourth leaf, starting from the apical leaf. Figure 4 shows the averaged photosynthetic efficiency for each group.

The polyphasic curves qualitatively show the photosystems’ correct functioning. Figure 4′s OJIP curves were plotted on a logarithmic axis to observe chlorophyll fluorescence over time, from minimum fluorescence (F0) up to maximum fluorescence (Fm). The treatments’ average curves almost overlap, indicating that there were no differences in the different groups’ photosystem II (PSII) function.

Regarding photosynthetic parameters, the photosynthetic efficiency Fv/Fm was always >0.8, and the performance index was always higher than one and reached a maximum of 12. High PI values indicated potential capacity for energy conservation from photons absorbed during PSII to the reduction of electron acceptors in the intersystem between PSII and PSI. A Student’s T-test confirmed that each sample’s photosynthetic apparatus worked regardless of the treatment applied and did not reveal significant differences in the Fv/Fm between the control groups and treated groups (*p* > 0.05); this confirmed the OJIP results. 

## 3. Discussion

Inoculation with PGPM and their consortia can help plants alleviate metal stress in metal-rich soils [16,21,32], counteracting physiological changes that cause lower biomass production and alterations in physiological responses, biochemical activities, and photosynthetic efficiency [5,6,7,8,43]. 

*Alyssoides utriculata* inoculated with single and mixed PGP inocula showed a remarkable increase in dry aboveground biomass (up to 4–5-fold over the control) in relation to more efficient phosphorous nutrition and the alleviation of metal stress, as already documented in recent studies regarding the beneficial effects of inoculated microbiota [16,21,32,44,45,46]. Such high biomass values support the effectiveness of PGPB *P. fluorescens* SERP1 to help plants cope with metal-rich soil stress and reach a significantly increased biomass compared with those reported in the literature. 

In fact, most papers about crops inoculated with other *P. fluorescens* strains (such as DR397, P22, *Pseudomonas* sp. Z6, and *P. fluorescens* biotype G) [35,36,47] identify an increase in shoot dry biomass 2–4 times lower than our findings, even if higher plant yields were obtained.

Similarly, the PGPF *Penicillium ochrochloron* used in this study showed properties similar to those of other PGP *Penicillium* strains, and was able to improve the dry biomass of horticultural plants (e.g., *Brassica campestris, Saccharum officinarum, Cucumis sativus*, etc.) from 0.5 up to 1.7-fold [44,48]. *Penicillium ochrochloron’s* indole acetic acid (IAA) and siderophore production, as well as its P solubilization, improved nutrient availability [48,49]. Interestingly, Tarroum et al. [50] recently demonstrated the direct (via inoculum) and indirect (using a cell-free culture filtrate) PGP activity of *Penicillium olsonii* in tobacco by increasing the dry biomass up to 4.8-fold compared to control plants; these results are similar to ours. Moreover, the PGPF effect significantly increased plant growth and total chlorophyll content. 

The bacterial and fungal mix used (*P. fluorescens* and *P. ochrochloron*) improved the aboveground biomass, as similarly documented for different microbial consortia used in crops, in *Citrus* sp. [36,51,52,53], and in the hyperaccumulator *Miscanthus* × *giganteus* (Mxg) J.M.Greef, Deuter ex Hodk., Renvoize [54]. Nevertheless, the documented consortia showed a lower increase in biomass (up to 3.1-fold) compared to our results. However, using our mixture did not have more effective results than single strains in terms of biomass increase (fivefold) (even if the increase was more relevant than those reported in the cited studies), confirming this consortium’s strong PGP ability. 

Regarding metal accumulation, a peculiar trend was noted for Mn, which increased twofold with fungi and bacteria treatments and fourfold with the mixed treatment. Few data are available to justify the increase in this nutrient (a minor and stable element of the tested soil) [12], even if the ability of some fungi to solubilize Mn from insoluble Mn oxides has already been documented [55]. Interestingly, bacteria and fungi inoculation supported a Zn increase; a similar ability was recorded in fungi for ericoid mycorrhizal strains that increased plants’ ability to solubilize inorganic Zn compounds, thus allowing better colonization of soils polluted with toxic metals [56]. Additionally, other plants inoculated with *Pseudomonas fluorescens* strains showed the same increases in Zn and Cu and consequent plant growth [57,58,59], which supports our current findings. In addition, the inoculation of sunflowers with *P. fluorescens* P22 and *Pseudomonas* sp. Z6 led to an increase in Zn and Mn uptake [47] that was two times lower than our results.

Unexpectedly, the mixed treatment’s copper accumulation in aboveground biomass was twofold greater than in the controls. This was an uncommon result, as both single bacterial and fungal PGP strains usually alleviate copper toxicity [55], reducing its accumulation in plant tissues [54], in contrast to our results. However, other studies documented increased accumulations of P, Zn, Cu, and Mn resulting from inoculation with other microbial consortia [51] or Zn accumulation in leaves and stems that increased a plant’s dry weight [54]. We cannot exclude that the synergistic effect of the two strains in the mixture might have left more copper available for plants because of an antagonistic competition for Cu between the two strains, as shown in other studies [60], further studies are required to clarify this response. 

The photosynthetic activity and global ecophysiological performance of plants in response to metals, usually employed to detect metal stress [5,6,7,8], demonstrated a stable Fv/Fm and plant performance in *Alyssoides utriculata* grown in a metal-rich soil. This contrasts with observations regarding other species inoculated with the bacterium *Pseudomonas fluorescens,* such as the plant *Sedum alfredii*, a Cd hyperaccumulator, that has shown a higher Fv/Fm value [61]. However, in a previous study conducted by Roccotiello et al. [62], increasing the Ni level resulted in a stable Fv/Fm among Ni treatments (always ≥0.8) and a PI >1.5. Our findings showed higher values of both parameters, highlighting the ability of this species to cope with other metals and avoid severe damage to its photosynthetic apparatus. A similar response has been demonstrated in species such as *Arundo donax* under Se treatments, where no alterations in the Fv/Fm ratio were observed in most of the selected ecotypes [63]. The response of *A. utriculata* confirms the constitutive ability of this species to cope with metals when living on metalliferous soils (such as the soil tested in our study), which is similar to other hyperaccumulators [64,65,66,67], and illustrates the peculiar morpho-physiological and biochemical adaptations typical of hyperaccumulator plants [68,69]. Interestingly, the bacteria and fungi tested in this study (and their combination) did not result in a better ecophysiological performance, as their main effects were related to the optimization of the plants’ mineral nutrition. Further investigations are needed to clarify this response. 

## 4. Materials and Methods

### 4.1. Plant Species and Soil Collection

The facultative Ni-hyperaccumulator *Alyssoides utriculata* (L.) Medik. (Figure 5) is a small thermophilic, xerophilous, suffruticose, and chamaephyte plant. It is found in the northeastern Mediterranean region; in Italy, it is quite commonly found in the Piedmont and Liguria regions, where it mainly grows on serpentinites [61].

*A. utriculata* plants were grown from seeds collected according to international guidelines [70]. Samples were harvested from the Ligurian Appenines. The presence of Ni in the mother plants was assessed using a colorimetric field Dimethylglyoxime (DMG) test.

Voucher herbarium specimens of *Alyssoides utriculata* plants and seeds were deposited in the University of Genoa herbarium, codes GE1904 and GE5364, respectively. The soil sampled in the ophiolitic Massif of the Voltri group (Beigua Geopark) originated from serpentinite bedrock, and its chemical analysis, as in [55], highlighted high metal concentrations, as follows: average nickel, (Ni) = 945 mg kg^−1^; magnesium (Mg), MgO = 21 wt%; manganese (Mn), MnOt = 0.20 wt%; Zinc, (Zn) = 150 mg kg^−1^; and copper, (Cu) = 25 mg kg^−1^. These concentrations were the starting conditions of our experiment. 

The soil was subjected to granulometric and mineralogical analyses and divided into 2 fractions, coarse (>4 mm) and medium-fine (<2 mm), equally mixed. To eliminate the original biotic component, the soil was sterilized via oven drying at 130 °C.

### 4.2. Experimental Design

The experimental design consisted of 80 samples, with 20 plant replicates for each treatment (control, bacteria, fungi, and mixed). After inoculation (see Section 4.3), pots were placed in completely randomized series. The trial lasted 18 months after inoculation. Each sample was labeled, classified with a unique number, and monitored over time via photographic documentation.

### 4.3. Soil Inoculation

The bacterial strain *Pseudomonas fluorescens* Migula, 1895 (SERP1) and the fungal strain *Penicillium ochrochloron* Biourge (SERP03 S) were used in vivo as inocula. The strain sequences were submitted to the NCBI GenBank^®^ database under accession numbers MG661811 and MG850978, respectively. These PGP strains had previously demonstrated the ability to co-grow effectively [32].

Two weeks after setting up the soil, the selected microbiotic components were inoculated. Each vessel was inoculated with 1 mL of bacterial or fungal suspension (or a mixture of the two) at a concentration of 10^8^ colony-forming units (CFUs). 

The pots were divided into four groups: control (20 pots with native soil as is, non-sterile and not inoculated); fungi (20 pots with sterile native soil and *Penicillium ochrochloron* inoculum); bacteria (20 pots with sterile native soil and *Pseudomonas fluorescens* inoculum); and mixed (20 pots with sterile native soil and bacterial–fungal co-inocula).

The pots were irrigated once a week (120 mL in each pot) using an automatic system (MySolem System) to maintain a 70% water holding capacity (WHC) to guarantee microbial and fungal activity. 

### 4.4. Seeds Sowing

Eighty seeds were sown in each substrate. Germinators were sterilized using hydrogen peroxide before being sown into a mixture of soil of natural origin and sterile vermiculite in a 1:1 ratio. Seeds were sterilized using 10% sodium hypochlorite and washed three times with sterile distilled water. Germinators were placed in a protected culture to maintain high relative humidity and promote germination.

### 4.5. Transplants of A. utriculata 

After 2 months, plants were subjected to mild fertilization with N-P-K (10:20:10) fertilizer at ¼ strength (2 mL for each plant). 

After 3 months, *A. utriculata* seedlings with 2–3 pairs of leaves were transplanted into labelled pots (containing one of the four substrates) in the experimental greenhouse. 

### 4.6. Ecophysiological Response

Ten months after transplant, direct measurements verified photosynthetic efficiency, that is, the ratio of energy accumulated during photosynthesis chemical reactions to absorbed light energy. 

For the analysis, 240 in vivo measurements (20 sample plants per treatment group and control and 3 leaves per plant) were made using the Hansatech Instruments HANDY-PEA tool.

The sensor unit consisted of a series of three diodes emitting a 650 nm peak wavelength (which was readily absorbed by chlorophyll) at a maximum intensity of 3500 µmol m^−2^ s^−1^ on the sample’s surface.

Leaves were dark-adapted for 20 min, then submitted to a 1-s pulse of ultra-bright red radiation. The recorded parameters were F = F0, Fm, Fv/Fm, and PI.

F0 is the minimum fluorescence value, representing emissions from the excited chlorophyll molecules in the photosystem II’s antenna structure.Fm is the maximum fluorescence value, obtained after applying a saturation pulse to a dark-adapted leaf.Fv/Fm indicates the maximum quantum efficiency of the photosystem II, which indicates a plant’s photosynthetic performance.PI (performance index) indicates a sample’s viability.

### 4.7. Biomass Evaluation

Twelve months after transplanting, belowground and aboveground biomass production was evaluated; roots were separated from leaves, washed several times under tap water and subsequently three times under deionized water, and oven-dried at 60 °C for 48 h before weighing. 

### 4.8. Analysis of Elements 

To quantitatively evaluate the elements’ concentrations in plants, dried aboveground biomass was weighed, powdered, and sent to the Land Analysis and Vegetable Productions Laboratory, Sarzana, SP, Italy, for ICP-MS analysis.

### 4.9. Data Analysis

Data were processed using Minitab 15 Statistical Software.

A descriptive statistical analysis for each dataset, and subsequently an inferential statistic, was created by applying the Student’s T-test to compare the means of unpaired samples. Radar graphs highlighted similarities and anomalies among the different treatment groups.

Parametric analysis verified whether the average value of a distribution significantly differed from a certain reference value; this facilitated studying the experimental groups via comparison with the control group. The confidence interval was 95%. PEA Plus Application software was used to analyze samples’ OJIP polyphasic curves.

## 5. Conclusions

Single and mixed inocula of fungi and bacteria on *Alyssoides utriculata* facilitated a specific response to metal stress in terms of the accumulation of elements and a related increase in aboveground plant biomass.

The eco-physiological performance of *A. utriculata* was stable with single and mixed PGP inocula and comparable with the control. Plants’ aboveground biomass was between four and five times higher in single and mixed inoculated plants over controls due to increased P uptake. All treated plants reached the Ni hyperaccumulation threshold with no observable differences in Ni concentrations. All treatments resulted in a Mn increase, the mixed inocula promoted the uptake of Cu twofold, and single inocula increased the Zn concentrations. Even if these elements’ concentrations were far below the hyperaccumulation threshold, these results indicate the selected single and combined PGP strains’ abilities to help plants cope with metals in the substrate. 

This study, through selected microbial strains, found an application to improve the rhizosphere resilience when the available nutrients are depleted and help plants coping with metal stress. In addition, it provides new insight for using selected PGP strains for a sustainable assisted phytoremediation approach in metal-contaminated soils.

## Figures and Tables

**Figure 1 plants-12-00554-f001:**
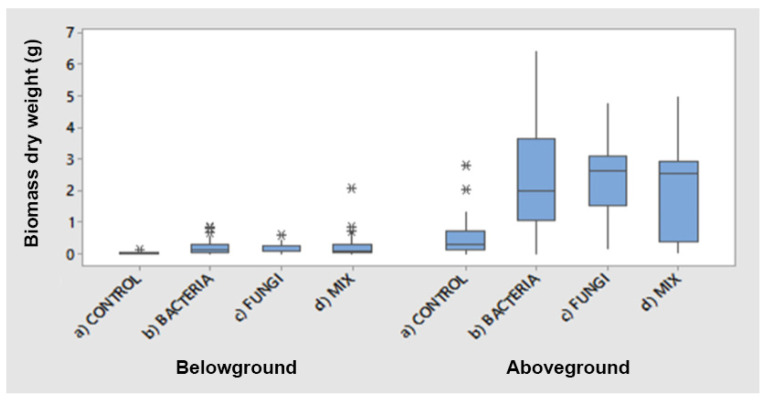
Boxplot of the belowground and aboveground dry biomass of different treatments (*n* = 20 for each treatment). The boxplot illustrates the interquartile range; central lines mark the data’s medians; box edges represent the first and third quartiles; whiskers show maximum and minimum values; and * represents the outliers.

**Figure 2 plants-12-00554-f002:**
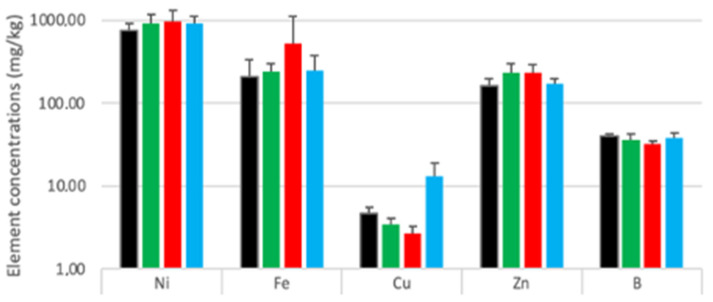
Micronutrients accumulation in the aboveground biomass associated with different treatments on a log scale. Data are average ± SD, *n* = 15 for each treatment, • control, • bacteria, • fungi, and • mixed.

**Figure 3 plants-12-00554-f003:**
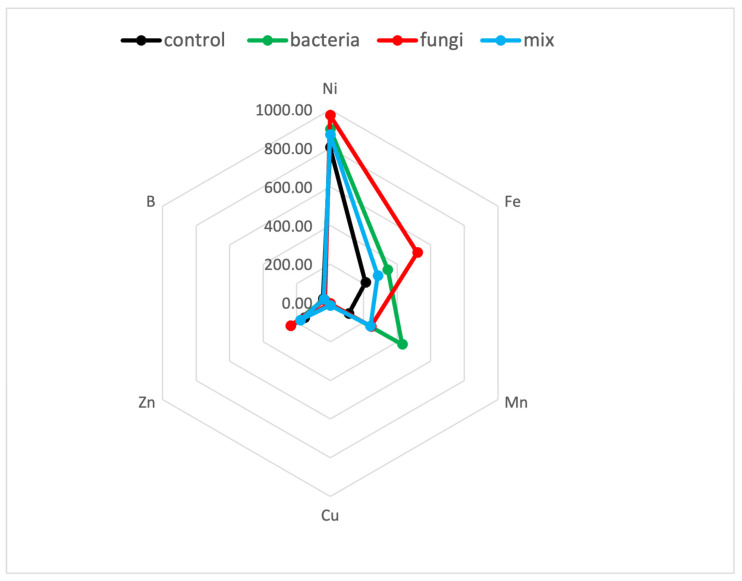
Radar charts without outliers showing the accumulation trends of the group of treatments with different accumulation in the dry aboveground biomass, expressed in mg/kg; *n* = 15, each group: • control, • bacteria, • fungi, • mix.

**Figure 4 plants-12-00554-f004:**
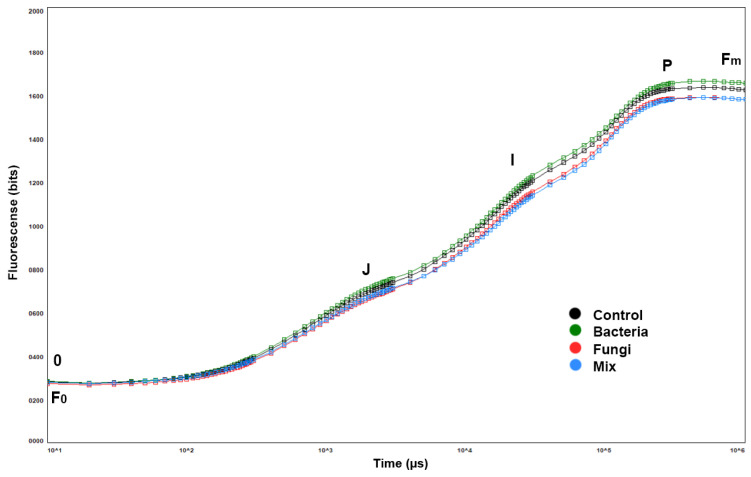
OJIP polyphasic curves; fluorescence transient analysis of *A. utriculata* treatments distinguished by curve colors; peaks are denoted by letters 0, J, I, and P, which correspond to fluorescence values measured at 50 ms (F0, step 0), 2 ms (step J), 30 ms (step I), and maximal (Fm, step P), respectively. Data are means of 60 measurements per treatment (20 plants in each treatment, 3 leaves per plant).

**Figure 5 plants-12-00554-f005:**
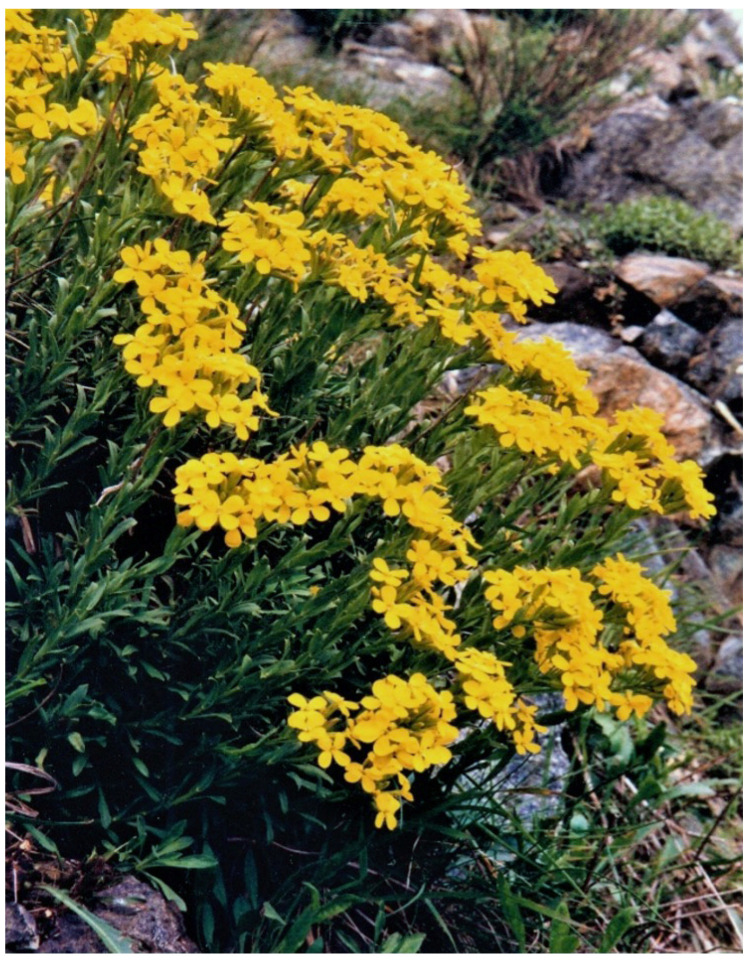
*Alyssoides utriculata* on serpentines (photograph by M. Calbi).

**Table 1 plants-12-00554-t001:** Student’s T-test results for different treatments’ dry belowground and aboveground biomass; significant: *p* < 0.05; highly significant: *p* < 0.001; and *n* = 20 for each treatment. Significant data are marked in bold, ns: not significant; superscript letters highlighted higher accumulation in controls (a) or in treatments (b).

Treatments	T-Test	
	Belowground	Aboveground
*Control* vs. *Bacteria*	**0.001 ^b^**	**0.000 ^b^**
*Control* vs. *Fungi*	**0.000 ^b^**	**0.000 ^b^**
*Control* vs. *Mix*	**0.010 ^b^**	**0.000 ^b^**
*Bacteria* vs. *Mix*	ns	ns
*Fungi* vs. *Mix*	ns	ns

**Table 2 plants-12-00554-t002:** Unpaired two-sample T-test results without outliers and related *p*-values of elements’ concentration in aboveground biomass. Significant data (*p* < 0.05) are marked in bold; ns: not significant; *n* = 15 for each treatment; superscript letters highlight higher accumulations in (a) controls or (b) treatments.

Treatments	*p* Value of Elements’ Concentration in the Aboveground Biomass											
	Ca	Mg	K	P	S	N	Ni	Fe	Mn	Cu	Zn	B
*Control* vs. *Bacteria*	ns	ns	ns	**0.000** **^b^**	**0.003** **^a^**	ns	ns	ns	**0.000** **^b^**	**0.003** **^a^**	**0.001** **^b^**	**0.015** **^a^**
*Control* vs. *Fungi*	ns	ns	ns	**0.016** **^b^**	**0.001** **^a^**	ns	ns	ns	**0.002** **^b^**	**0.001** **^a^**	**0.003** **^b^**	**0.001** **^a^**
*Control* vs. *Mix*	ns	ns	ns	**0.000** **^b^**	**0.002** **^a^**	**0.024** **^a^**	ns	ns	**0.001** **^b^**	**0.001** **^b^**	ns	ns

## Data Availability

Not applicable.

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
