# Peer review of "Effects of Bacterial and Fungal Inocula on Biomass, Ecophysiology, and Uptake of Metals of Alyssoides utriculata (L.) Medik."

_plants, 2023, doi:10.3390/plants12030554_

Round 1
Reviewer 1 Report
The manuscript required moderate revision.
The title is to be as
Individual and combined effects of bacterial and fungal inocula on biomass, ecophysiology, and uptake of metal on Alyssoides utriculata (L.) Medik.
Line 17 of the abstract removed "randomized".
Line 19 it is ICP-MS or IC-MS, I think. Please cross-check.
Fig is not required as it is well known.
Mention full form first the word appears like in line 71 in place of 75-76
Remove lines 82-83. Elaborate on the results.
Results require elaboration. Only by seeing the figures and tables, everything is not clear.
All the results are not properly discussed. Correct and correlate the results with earlier findings then infer the conclusion.
Author Response
We are thankful to the reviewer for the encouraging comments on our manuscript.
The manuscript was substantially changed in the Abstract and Introduction section, where more pertinent citations were added. The Discussion and the Conclusion were totally rewritten, better supporting our main findings. In the Results section we better described each main finding to better support figure and tables. In the Material and methods section, we provide additional details on sample design, data analysis and native soil characteristics.
Please, find below our point-by-point answers.
The title is to be as
Individual and combined effects of bacterial and fungal inocula on biomass, ecophysiology, and uptake of metal on Alyssoides utriculata (L.) Medik.
Thanks for the suggestion. We decided to change the title to “Effects of bacterial and fungal inocula on biomass, ecophysiology, and uptake of metals of Alyssoides utriculata (L.) Medik.”
Line 17 of the abstract removed "randomized".
Thank you. The abstract was almost completely re-written. We removed “randomized” from the text.
Line 19 it is ICP-MS or IC-MS, I think. Please cross-check.
Thanks. Text was carefully checked for language. All typing errors like the one mentioned were corrected.
Fig is not required as it is well known.
Suggestion taken. We removed figure 1 from the manuscript.
Mention full form first the word appears like in line 71 in place of 75-76
Thank you. We carefully checked and corrected these lines and throughout the manuscript.
Remove lines 82-83. Elaborate on the results.
Done, thank you. The Results were better elaborated.
Results require elaboration. Only by seeing the figures and tables, everything is not clear.
All the results are not properly discussed. Correct and correlate the results with earlier findings then infer the conclusion.
Thanks for the comment. In the Results section we better described each main finding to better support figure and tables. Discussion and Conclusion were restructured and rewritten on the basis of such results.
Reviewer 2 Report
The article expatiates the role of Nickel hyperaccumulator Alyssoides utriculata (L.) Medik. Nickel hyperaccumulator Alyssoides utriculata (L.) Medik. response to bacterial and fungal inocula: biomass, ecophysiology, and uptake.
English should improve by a native person. The paper suffers from a poor English structure throughout and cannot be published or reviewed properly in the current format. The manuscript requires thorough proofreading by a native person whose first language is English. The instances of the problem are numerous and this reviewer cannot individually mention them. It is the responsibility of the author(s) to present their work in an acceptable format. Unless the paper is in a reasonable format, it should not have been submitted.
The scientific background of the topic is poor.
Overall, the abstract looks very vague and it does not add much to our current understanding of the responses to the nickel hyperaccumulation effect
The introduction should be improved by highlighting more cited works related to the topic of the article.
Numerous grammatical and word choice errors.
Scientific writing is not just writing about science; it is the technical writing that scientists do to communicate their research to others In this manuscript non-scientific wording is used for example "Even if Zn increased thanks to the treatments with bacteria".
Figure 1 is out of context, and not related to the manuscript.
There is no repetition of the experiment as repeating multiple trials in an experiment helps to reduce the effect of errors.
Experiments should be repeated twice, and more analyses are required
The amount of heavy metals present in the soil before and after experimentation is not clearly mentioned
The experimental material voucher number is missing.
The design of the experiment is not appropriate, and more information is required for the replication of the experiment's treatments
The chemistry of substrate deriving from serpentinite mother rock is also missing
Author Response
We are thankful to the reviewer for the comments on our manuscript and we are sorry for the impression given, despite of the high amount of work done.
The manuscript was substantially changed in the Abstract and Introduction section, where more pertinent citations were added. The Discussion and the Conclusion were totally rewritten, better supporting our main findings. In the Results section we better described each main finding to better support figure and tables. In the Material and methods section, we provide additional details on sample design, data analysis and native soil characteristics.
Please, find below our point-by-point answers.
English should improve by a native person. The paper suffers from a poor English structure throughout and cannot be published or reviewed properly in the current format. The manuscript requires thorough proofreading by a native person whose first language is English. The instances of the problem are numerous and this reviewer cannot individually mention them. It is the responsibility of the author(s) to present their work in an acceptable format. Unless the paper is in a reasonable format, it should not have been submitted.
Thank you. Most of the manuscript sections like abstract, introduction, discussion and conclusion were reorganized and rewritten, and the final text was submitted for English check, as per reviewer suggestion.
The scientific background of the topic is poor.
Thank you for the comment. We added literature to provide a better outline of the current state of knowledge. Consequently, the introduction was restructured, and more literature added for a comprehensive overview. The same for the Discussion section, that was totally rewritten and where we add more than 50% new literature to better support our main findings
Overall, the abstract looks very vague and it does not add much to our current understanding of the responses to the nickel hyperaccumulation effect
Thank you. As previously mentioned, the abstract was rewritten to clearly highlight our main findings and especially to better support how PGPM and their consortia can change the response of Ni-hyperaccumulator plants under metal stress.
The introduction should be improved by highlighting more cited works related to the topic of the article.
The introduction was reorganized and partly rewritten, adding about 50% more pertinent papers. Most of them were published in these last 5 years or are highly relevant to the topic.
Numerous grammatical and word choice errors.
Scientific writing is not just writing about science; it is the technical writing that scientists do to communicate their research to others. In this manuscript non-scientific wording is used for example "Even if Zn increased thanks to the treatments with bacteria".
We are sorry for the poor language quality of the previous version of the manuscript. The English language was carefully checked and corrected throughout the text, as per reviewer suggestion. We hope that the current version will suit the quality standard of the journal and the correctness expected by the reviewer.
Figure 1 is out of context, and not related to the manuscript.
Thank you. We removed figure 1 since useless.
There is no repetition of the experiment as repeating multiple trials in an experiment helps to reduce the effect of errors.
Experiments should be repeated twice, and more analyses are required
The experimental design consisted in a total number of samples n=80, with 20 plant replicates for each treatment: control, bacteria, fungi, and mix, respectively. After inoculation, pots were completely randomized. The trial lasted 18 months from the inoculation. Each sample was labeled and classified with a unique number and monitored with photographic documentation over time.
The number of replicates is sufficient to apply the type of inferential statistical analysis chosen and to reduce random errors, both for biomass and for eco-physiological parameters.
This experiment is a long-time trial study, necessary for the development of dynamics related to PGP microorganisms in relations with plant roots. In the examined literature, studies generally lasted for shorter time, while for microbial dynamics the time factor is of a key importance. This experiment is part of a sequential trial, and it represents the step prior to the field test of the selected plant inoculated with bacterial and fungal strains. The volume of soil tested is significant but not comparable with field and related dynamics. In the field, trials errors due to environmental factors, that vary over time and in space, typical of the natural environment, cannot be considered comparable with controlled mesocosm environment.
We assumed that the field test will be affordable and will help providing additional new insight into the current state of knowledge, reducing the effects of errors, since the overarching goal of our study is providing microbial consortia for sustainable assisted-phytoremediation.
The amount of heavy metals present in the soil before and after experimentation is not clearly mentioned
We added details on the concentration of soil elements at the beginning of the experiment in the materials and methods section. Since the objective of the work is focused on how plants can vary their ability to accumulate elements under metal stress, metals and other elements were evaluated in the plants, focusing on the harvestable fraction, i.e., aboveground biomass.
The experimental material voucher number is missing.
Thank you. Voucher herbarium specimens of Alyssoides utriculata plant and seeds were deposited in the herbarium of the University of Genoa code: GE1904, and GE 5364, respectively.
The bacterial and fungal strains employed for this study were Pseudomonas fluorescens Migula 1895 (SERP1) and Penicillium ochrochloron Biourge (SERP03S). Both strain sequences were submitted in the NCBI GenBank® database under the accession number MG661811 and MG850978, respectively (Rosatto et al. 2019).
We added this information in the material and methods section.
The design of the experiment is not appropriate, and more information is required for the replication of the experiment's treatments
Thank you for the comment. Please, see our response to your previous comment on the experimental design and replicates.
The chemistry of substrate deriving from serpentinite mother rock is also missing
We added details on the concentration of soil elements in the materials and methods section.
Round 2
Reviewer 2 Report
Use experimental conducting pictures with appropriate labelling for treatments
Use alphabets for comparison of treatment means for all the parameters
The English language is so weak and many grammatical mistakes are throughout the text.
Author Response
We are thankful to the reviewer for the useful comments on our manuscript.
We accepted all the suggestions and specifically:
Use experimental conducting pictures with appropriate labelling for treatments
We decided to delete figure 3 from the main text placing it as supplementary material. Table 1 was edited and presented according to the style of table 2.
Table 3 was deleted from the text since it did not show any significat values for the considered parameters, as described in the main text.
Each figure and table was double checked for appropriate labelling. Current figure 3 was edited to be comparable with the other colour labelling used.
Use alphabets for comparison of treatment means for all the parameters
Done, thank you.
The English language is so weak and many grammatical mistakes are throughout the text.
The text was checked and corrected by a mother-tongue speaker provided by the English editing MDPI service.